# Probiotic *Bifidobacterium longum* supplied with methimazole improved the thyroid function of Graves' disease patients through the gut-thyroid axis

Dongxue Huo[1,2,3], Chaoping Cen[1,3], Haibo Chang[1], Qianying Ou[1], Shuaiming Jiang[2], Yonggui Pan[2], Kaining Chen[1✉] & Jiachao Zhang [1,2✉]

Graves' disease (GD) is an autoimmune disorder that frequently results in hyperthyroidism and other symptoms. Here, we designed a 6-month study with patients divided into three treatment groups, namely, methimazole (MI, $n = 8$), MI + black bean ($n = 9$) and MI + probiotic *Bifidobacterium longum* ($n = 9$), to evaluate the curative effects of probiotics supplied with MI on thyroid function of patients with GD through clinical index determination and intestinal microbiota metagenomic sequencing. Unsurprisingly, MI intake significantly improved several thyroid indexes but not the most important thyrotropin receptor antibody (TRAb), which is an indicator of the GD recurrence rate. Furthermore, we observed a dramatic response of indigenous microbiota to MI intake, which was reflected in the ecological and evolutionary scale of the intestinal microbiota. In contrast, we did not observe any significant changes in the microbiome in the MI + black bean group. Similarly, the clinical thyroid indexes of patients with GD in the probiotic supplied with MI treatment group continued to improve. Dramatically, the concentration of TRAb recovered to the healthy level. Further mechanistic exploration implied that the consumed probiotic regulated the intestinal microbiota and metabolites. These metabolites impacted neurotransmitter and blood trace elements through the gut-brain axis and gut-thyroid axis, which finally improved the host's thyroid function.

[1] Department of Endocrinology, Hainan General Hospital, School of Food Science and Engineering, Hainan University, Haikou, China. [2] Key Laboratory of Food Nutrition and Functional Food of Hainan Province, Haikou, China. [3] These authors contributed equally: Dongxue Huo, Chaoping Cen. ✉email: kainch@sina.com; zhjch321123@163.com

Graves' disease (GD), the most common systemic autoimmune disorder, affects 2–5% of the population[1] and peaks in incidence among patients aged 30–50 years[2,3]. GD has a higher risk prevalence of 3% in females compared with 0.5% in males during their lifetime[3–5], with 20–50 annual cases per 100,000 individuals[6]. The most common symptoms of GD include weight loss, fatigue, heat intolerance, tremor, and palpitations[3]. Ophthalmopathy is a serious extrathyroidal symptom that can threaten sight in 3–5% of patients.

Currently, one of the treatments for Graves' hyperthyroidism is to reduce TH synthesis by using antithyroid drugs (ATDs)[7], including propylthiouracil (PTU), carbimazole (CBZ), and methimazole (MI). MMI is recommended as the preferred thionamide drug for every nonpregnant patient with newly diagnosed Graves' hyperthyroidism for 18–36 months[2]. Indeed, patients with persistently high thyroid-stimulating hormone receptor autoantibodies (TSH-R-Ab, the specific biomarker for GD) at 12–18 months still require continued MMI therapy. Notably, the side effects of ATD are a skin rash, urticaria, increased risk for hepatitis and cholestasis[8], and major side effects occur in 0.2–0.3% of patients[9,10]. Even with persistent medication, the discontinuation of MMI and PTU is strongly associated with a high risk of recurrence[3]. Therefore, the disadvantages of ATD are the high recurrence rate and frequent testing required[11].

Graves' disease is a consumptive disease, so the patient should be arranged a nutrient-rich, easily digestible diet, supplemented with enough calories, nutrients, and vitamins to correct the consumption caused by the disease. In traditional Chinese medicine, black beans can be used as a dietary prescription for hyperthyroidism alone, which has a good effect on improving the symptoms of hyperthyroidism patients, such as postillness weakness and excessive sweating. At the same time, black beans contain rich protein, fat, carbohydrates, vitamins, and a variety of minerals. Black bean has high energy and is easy to digest, which is of great significance to meet the metabolic consumption needs of patients with Graves' disease. Growing evidence indicates that the intestinal microbiota is closely related to various autoimmune diseases, including primary hypothyroidism[12], type 2 diabetes[13], inflammatory bowel diseases[14], obesity[15], and rheumatoid arthritis[16]. Probiotics are live microorganisms that can potentiate health benefits on hosts when administered in adequate amounts[17], in general containing *Bifidobacterium* spp. and *Lactobacillus* spp., which are well known for their role in regulating and rebuilding the host's gut microbiome, and microbiota-targeted therapy using probiotics has been used to prevent and treat a variety of metabolic diseases, including type 2 diabetes[18], polycystic ovary syndrome[19], and hyperglycaemia[20]. However, the benefit of microbiota-targeted therapy for GD with probiotics appears dubious[1], even though the data suggested that patients on probiotics were significantly less likely to have hyperthyroid relapse[21], and they have a positive effect on trace elements such as iron, zinc, and copper[22].

To address the issues mentioned above, the effectiveness of probiotics in the treatment of GD by modulating the intestinal microbiota and the high recurrence rate in the treatment of GD, we designed a 6-month study with 25 patients with GD using three treatments: methimazole (MI, group A), MI + black bean (group B) and MI + probiotic (group C) (Fig. 1). All subjects stayed on their treatment for 6 months. We longitudinally tracked the blood and thyroid indexes every month, and the gut microbiome was determined by shotgun metagenome sequencing. Short-chain fatty acids (SCFAs) and trace elements were evaluated at baseline, 3 and 6 months during the 6-month treatment period. The present study extended our understanding of the mechanism of the probiotic *Bifidobacterium longum* supplied with MMI in regulating the gut microbiota and reducing the recurrence rate in patients with GD and also developed a potential useful therapy for Graves' disease based on the intestinal microbiota regulation.

## Results

**Methimazole intake altered the intestinal microbiota in ecological and evolutionary aspects while improving thyroid function in patients with GD.** Unsurprisingly, methimazole intake significantly improved the thyroid indexes, including FT3, FT4, and TSH, after taking the drug for just 1 month (Fig. 2a).

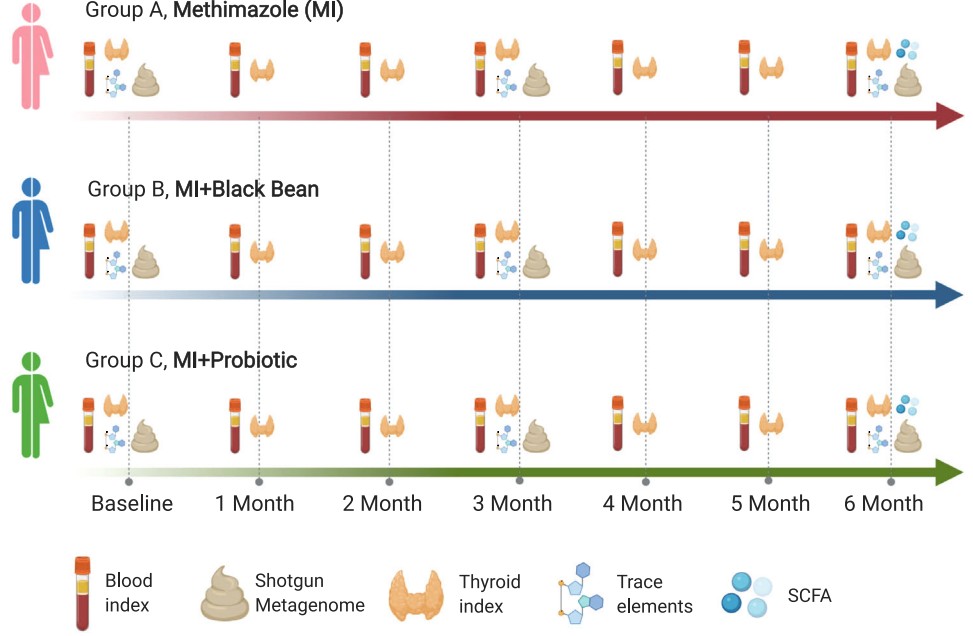

**Fig. 1 Experimental design and sampling time points design.** We longitudinally tracked blood and thyroid index at every month, gut microbiome determined by shotgun metagenome sequencing, short-chain fatty acids (SCFAs) and trace elements were evaluated at baseline, 3 and 6 months during the 6-month treatment.

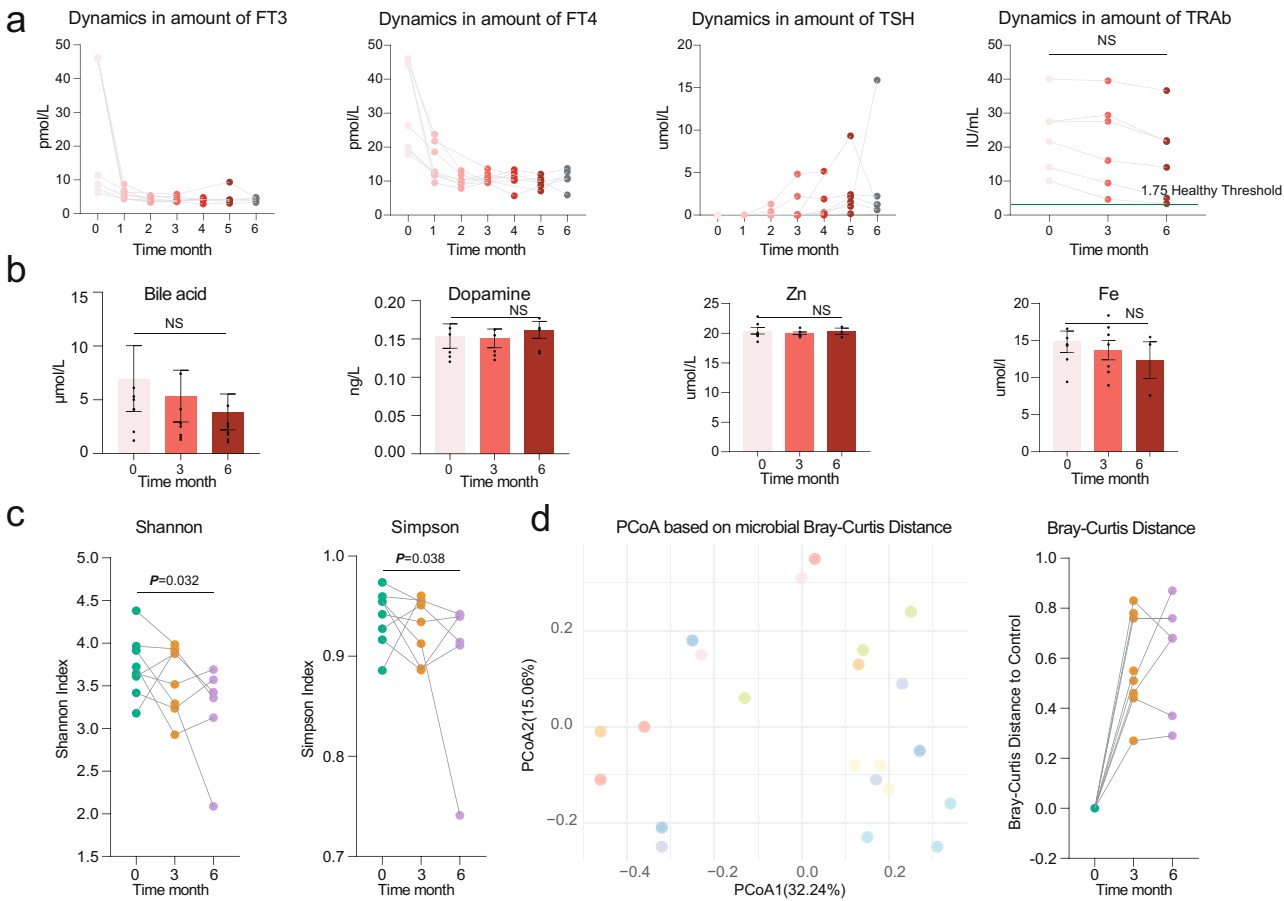

**Fig. 2 The methimazole intake improved the thyroid function of GD patients. a** Dynamic in the concentration of thyroid indexes including free triiodothyronine (FT3), free thyroxine (FT4), thyroid-stimulating hormone (TSH), and thyroid-stimulating hormone receptor antibodies (TRAb) during the whole experiment, these indexes were determined monthly. **b** Dynamic in the concentration of bloody bile acid, Dopamine, Zn, and Fe, these indexes were determined at baseline, month 3 and month 6. **c** The impacts of methimazole on intestinal microbial alpha diversity, the points in the same color represented the different subjects in the same time points. **d** Principal coordinates analysis (PCoA) based on Bray-Curtis distances of metagenomic species, the points in the same color represented the same subjects in the different time points.

Meanwhile, no significant change was observed in the concentrations of blood bile acid, dopamine, and Fe and Zn between the samples comparing the baseline and at the end of treatment (Fig. 2b). However, the average amounts of thyrotropin receptor antibody (TRAb) at month six were still well above the healthy control level (1.75 IU/mL), which indicated a high GD recurrence rate in this treatment group (Fig. 2a).

We next sought to explore the impacts of methimazole intake on the intestinal microbiome of patients with GD. The Shannon and Simpson indexes, which represented the microbial alpha diversity, were calculated based on the metagenomic species profile. Then, a sharp decline was observed in the microbial Shannon index between the baseline samples and the samples treated for 6 months with methimazole (Fig. 2c). We constructed PCoA ordinations based on the Bray-Curtis distance (Fig. 2d) among the taxonomic profiles and calculated the microbial Bray-Curtis distance of each subject from the baseline to each time point. Notably, methimazole intake significantly altered the intestinal microbial structure, although individual heterogeneity could not be ignored (Fig. 2d). Accordingly, we identified species with significant differences among the three-time points. Specifically, *Faecalibacterium prausnitzii*, *Ligilactobacillus salivarius*, *Lactococcus lactis*, and some species of the genera *Porphyromonas* and *Prevotella* significantly decreased (Fig. 3a). We further assembled the metagenomic reads into contigs and constructed

metagenomic assembled genomes (MAGs) in each subject (Fig. 3b). Then, we identified MAGs with significantly differential abundances among the three-time points and constructed a violin figure with represented MAGs (Fig. 3c). Phylogenomic analysis of the MAGs suggested overall consistency with the taxonomic annotation. The MAGs of *Faecalimonas nexilis*, *Erysipelatoclostridium ramosum* and *Anaerobutyricum hallii* continued to grow, while the MAGs of *Prevotella* sp. significantly decreased.

After demonstrating the disorder in the intestinal microbiota, we further explored the changes in microbial metabolic pathways between the baseline samples and the samples treated with 6 months of methimazole (Fig. 3d). Annotated by the UniRef protein database, we obtained profiles of microbial gene families and metabolic pathways. Among the differentially abundant metabolic pathways, xylose degradation, L − phenylalanine degradation, and L − ascorbate biosynthesis significantly increased in the samples with 6 months of methimazole treatment, whereas the microbial metabolic abilities of tetrapyrrole biosynthesis, peptidoglycan biosynthesis, and lactose and galactose degradation decreased significantly.

Beyond the taxonomic and functional features, we further explored the evolutionary changes at the genetic level in intestinal microbial species that responded to methimazole intake. We aligned the metagenomic data against the reference genomes of species with relative abundances higher than 0.5% in the subjects of group A at different time points and reconstructed a profile of

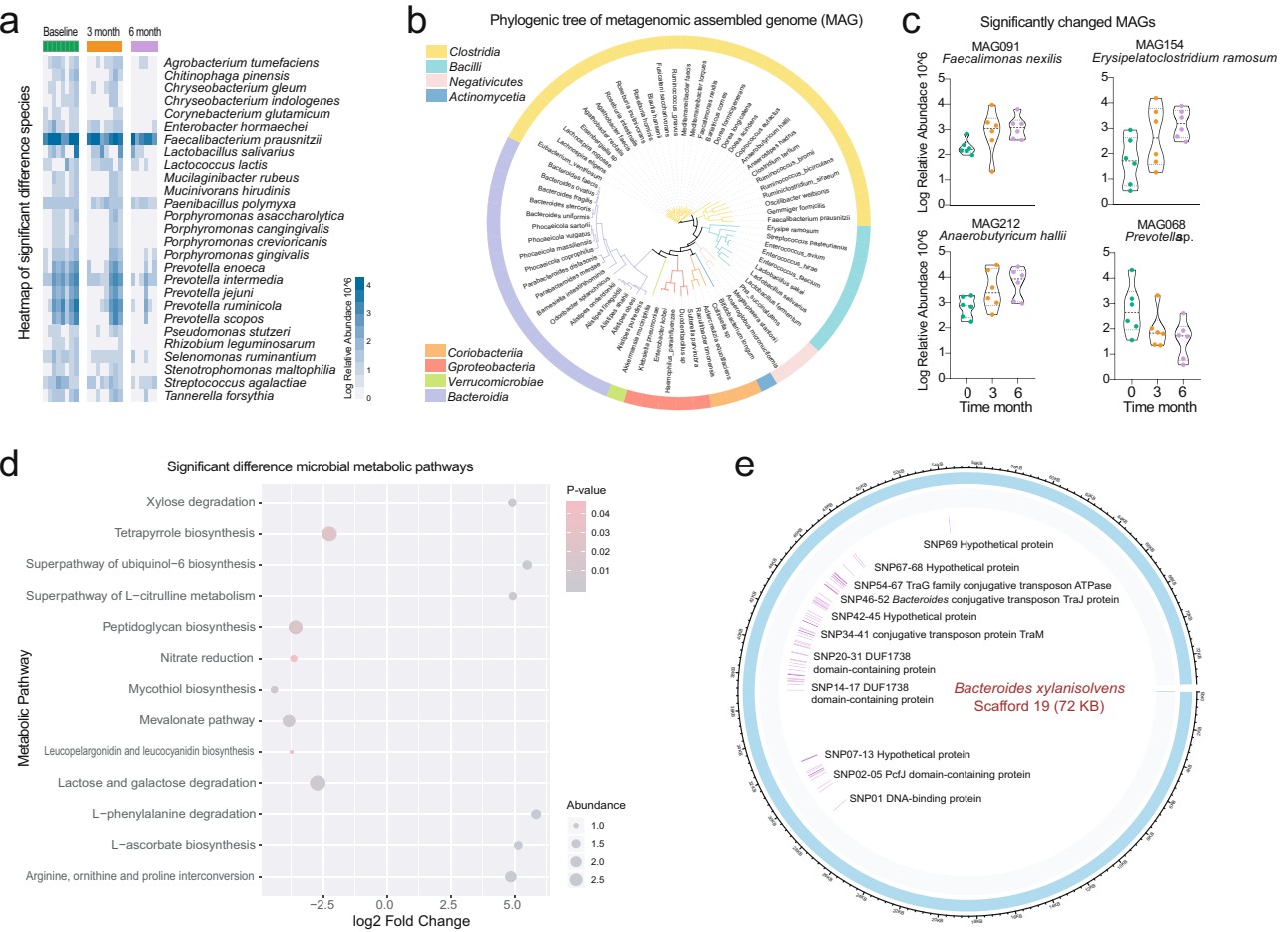

**Fig. 3 The methimazole intake alerted the intestinal microbiota in the ecological and evolutionary aspect. a** The significantly different metagenomic species among different time points. **b** Phylogenetic tree of the MAGs with clades colored by order. **c** MAGs of significant difference among different time points. **d** The intestinal microbial metabolic pathway that differed significantly between the samples at baseline and the samples with 6 months of methimazole treatment. **e** Genomic locations and contexts of SNPs in the species of *Bacteroides xylanisolvens*. These SNPs in *Bacteroides xylanisolvens* genome are mainly enriched in scaffold 19 and these genes related to the function of DNA-binding protein, conjugative transposon protein, and TraG family conjugative transposon ATPase.

SNPs (a SNP was identified and confirmed only when the following occurred: i. the mutation was detected compared with the original base of the microbial genome at baseline; ii. the mutation was detectable in the following two time points; iii. the quantity score for each annotated SNP was more than 60). A total of 77 SNPs were annotated in six common intestinal species, with the number of SNPs ranging from 1 to 69. Among them, 90% of SNPs ($n = 69$) were annotated in the species *Bacteroides xylanisolvens* (Fig. 3e). These SNPs in the *Bacteroides xylanisolvens* genome were mainly enriched in scaffold 19, and these genes were related to the functions of DNA-binding proteins, conjugative transposon proteins, and TraG family conjugative transposon ATPases (Fig. 3e).

**Black-bean adjuvant methimazole intake maintained intestinal microbiome homeostasis in patients with GD during the 6-month treatment**. For patients in the MI + black bean treatment group (group B), we observed a similar curative effect in thyroid functional indexes and blood trace elements as that in the methimazole treatment group (Fig. 4a, b). The FT3, FT4, TSH, and bile acid significantly decreased after receiving the corresponding treatment for just 1 month, while the concentration of TRAb at month 6 was still higher than the healthy control level. Since the clinical indexes between the two groups above were similar, we

further explored the indigenous intestinal microbiome of patients with GD in group B who responded to methimazole and black bean treatment. Interestingly, the colour points representing the intestinal microbial structure in Fig. 5b clustered by the individual obviously, especially for the points at 3 and 6 months of the same volunteer (Fig. 5b), which indicated a temporary stable gut microbiota of subjects in group B after receiving the methimazole and black-bean treatment for 3 months. This observation was confirmed by dynamics in microbial alpha diversity, and no significant differences were found in the Shannon or Simpson indexes among samples at the last time points (Fig. 5a). Therefore, we claimed that methimazole treatment was able to sharply and persistently impact the intestinal microbiota of patients with GD, while supplementation with black beans for 3 months could buffer the drastic shock introduced by methimazole and maintain intestinal microbiome homeostasis. Even though the change in microbial structure was limited, we could also observe some specific changes at the microbial species level (the significantly changed species in group B were defined as follows: i. no significant change in the relative abundance of species at the baseline between the 2 groups; ii, no significant change in the relative abundance of species in group A during the entire experiment; iii, significant changes were observed in the relative abundance of species in subjects of group B at both 3 months and 6 months). The species *Bacillus litoralis*, *Streptococcus*

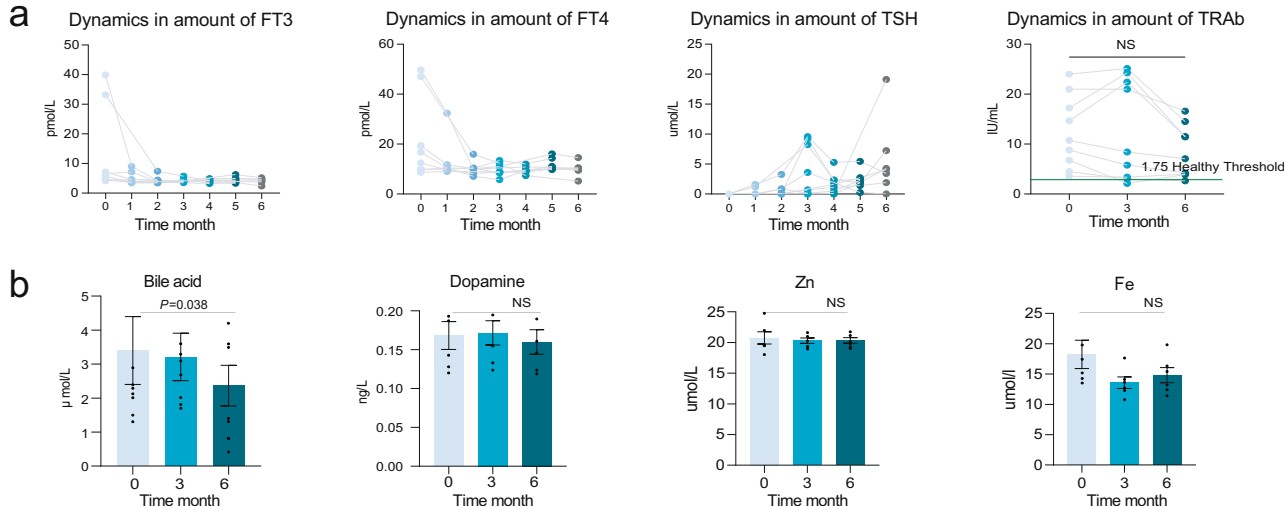

**Fig. 4 The black bean adjuvant methimazole intake improved the thyroid function of GD patients. a** Dynamic in the concentration of thyroid indexes including FT3, FT4, TSH, and TRAb during the whole experiment, these indexes were determined monthly. **b** Dynamic in the concentration of bloody bile acid, Dopamine, Zn, and Fe, these indexes were determined at baseline, month 3 and month 6.

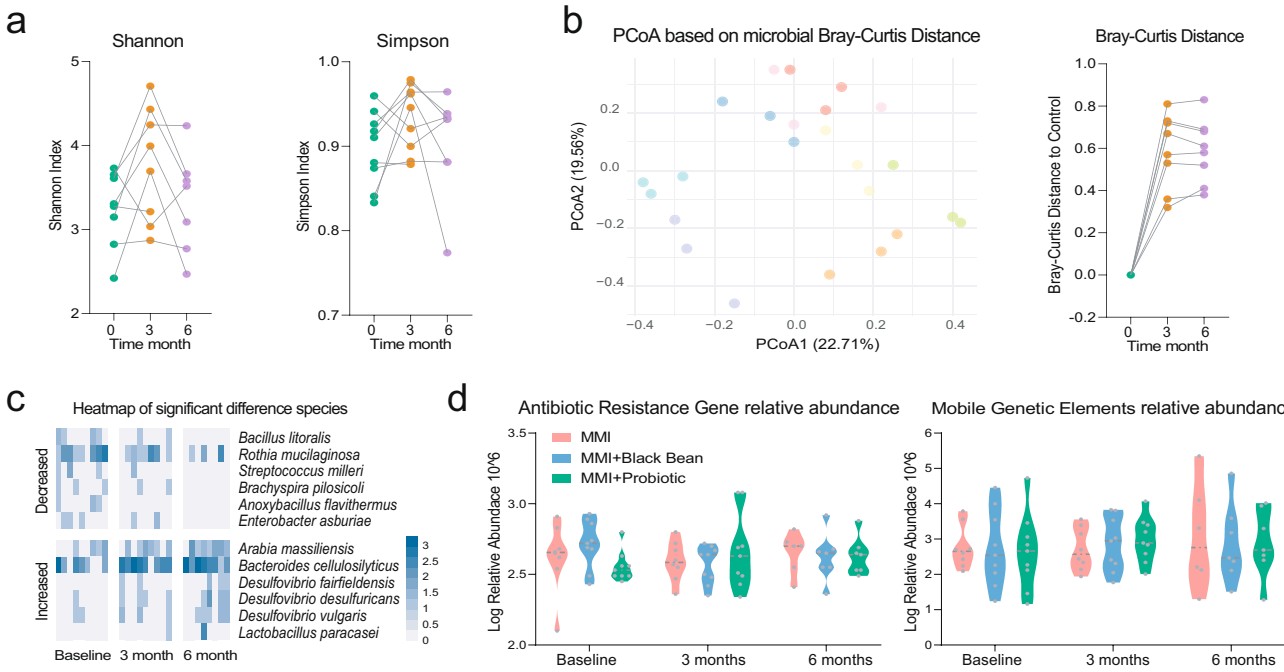

**Fig. 5 The black bean adjuvant methimazole intake maintained the intestinal microbiome homeostasis of GD patients during the 6-month treatment. a** The impacts of methimazole on intestinal microbial alpha diversity, the points in the same color represented the different subjects in the same time points. **b** Principal coordinates analysis (PCoA) based on Bray-Curtis distances of metagenomic species, the points in the same color represented the same subjects in the different time points. **c** The significantly different metagenomic species among different time points. **d** Dynamic in the relative abundance of function genes including antibiotic resistance gene and mobile genetic elements.

*milleri*, and *Enterobacter asburiae* decreased significantly, while some species of the genus *Desulfovibrio* increased gradually (Fig. 5c). However, we did not observe any significant changes in MAGs, antibiotic resistance genes, mobile genetic elements, microbial metabolic pathways, or microbial genome mutations (Fig. 5d), which further confirmed our deduction that supplementation with black beans for 3 months could maintain intestinal microbiome homeostasis for patients with GD when methimazole was consumed.

**Probiotic *Bifidobacterium longum* adjuvant methimazole treatment improved thyroid function and significantly reduced the TRAb concentration of patients with GD**. Similarly, the clinical thyroid indexes of patients with GD in the probiotic adjuvant methimazole treatment group (group C) continued to improve during the 6 months. FT3, FT4, TSH, dopamine, bile acid, and blood Fe significantly decreased at the end of the experiment (Fig. 6a, b). Amazingly, the concentration of TRAb, which is an indicator of the GD recurrence rate, recovered to the healthy control level.

Then, we sought to reveal the responses of indigenous gut microbiota to the probiotic *Bifidobacterium longum* and methimazole intake at the ecological and evolutionary levels. On the one hand, the probiotic supplied with methimazole treatment significantly impacted the patients' gut microbial structure, including microbial alpha and beta diversity (Fig. 6c). On the other hand, some species and MAGs, including *Bifidobacterium adolescentis*, *Bifidobacterium angulatum*, *Bifidobacterium breve*, *Bifidobacterium longum*, and *Faecalibacterium prausnitzii*, increased significantly, while *Blautia hansenii*, *Clostridium esthertheticum*, and *Klebsiella pneumoniae* decreased sharply (Fig. 6d and Fig. 7a). Meanwhile, the microbial metabolic pathways of fatty acid biosynthesis, toluene degradation, phenylacetate degradation, and flavin biosynthesis were enriched in subjects receiving the probiotic *Bifidobacterium longum* treatment for 6 months (Fig. 7b). Accordingly, propionic acid and butyric acid represented short-chain fatty acids that increased gradually (Fig. 7c). Furthermore, we uncovered the evolutionary changes at the genetic level in intestinal microbial species that responded to probiotic intake. A total of 12 SNPs were annotated in three genomes, with the number of SNPs ranging from 2 to 8. Among them, eight SNPs were annotated in the species of *Roseburia hominis* (Fig. S1), and most of them were synonymous mutations. Two SNPs (nonsynonymous mutations) were annotated in the *Bifidobacterium pseudocatenulatum* genome, and the mutated gene was related to the function of Ig-like domain-

containing protein (Fig. 3d). The other two SNPs (nonsynonymous mutations) were annotated in the *Bifidobacterium longum* genome, and the mutated gene was related to the function of the LPXTG cell wall anchor domain-containing protein (Fig. 3e).

After confirming that the probiotics were able to regulate the intestinal microbiome and improve thyroid function of patients with GD, we were eager to explore the potential mechanism underlying the interaction between the probiotics and thyroid clinical indexes. To address this question, we constructed a network including the probiotic *Bifidobacterium longum*, probiotic closely related metagenomic species ($r > 0.4$), SCFAs, dopamine, bile acid, blood trace elements, and thyroid functional indexes based on the determined Spearman's rank correlation coefficients (Fig. 8a). As shown in Fig. 8a, a generally positive correlation was observed between the probiotic *Bifidobacterium longum* and other *Bifidobacterium* species. These *Bifidobacterium* species were positively correlated with SCFAs but negatively correlated with bile acid, Fe, and dopamine. Furthermore, the positive correlations were found between the blood Fe and bile acid and thyroid FT3, FT4, and TRAb. Accordingly, we simplified a diagrammatic drawing, including the gut-brain axis and gut-thyroid axis to visualize the potential effective mechanism (Fig. 8b). Overall, the consumed probiotic regulated the intestinal microbiota and metabolites. These metabolites impacted neurotransmitter and blood trace elements through the gut-thyroid axis and gut-brain axis, which finally improved the host's thyroid

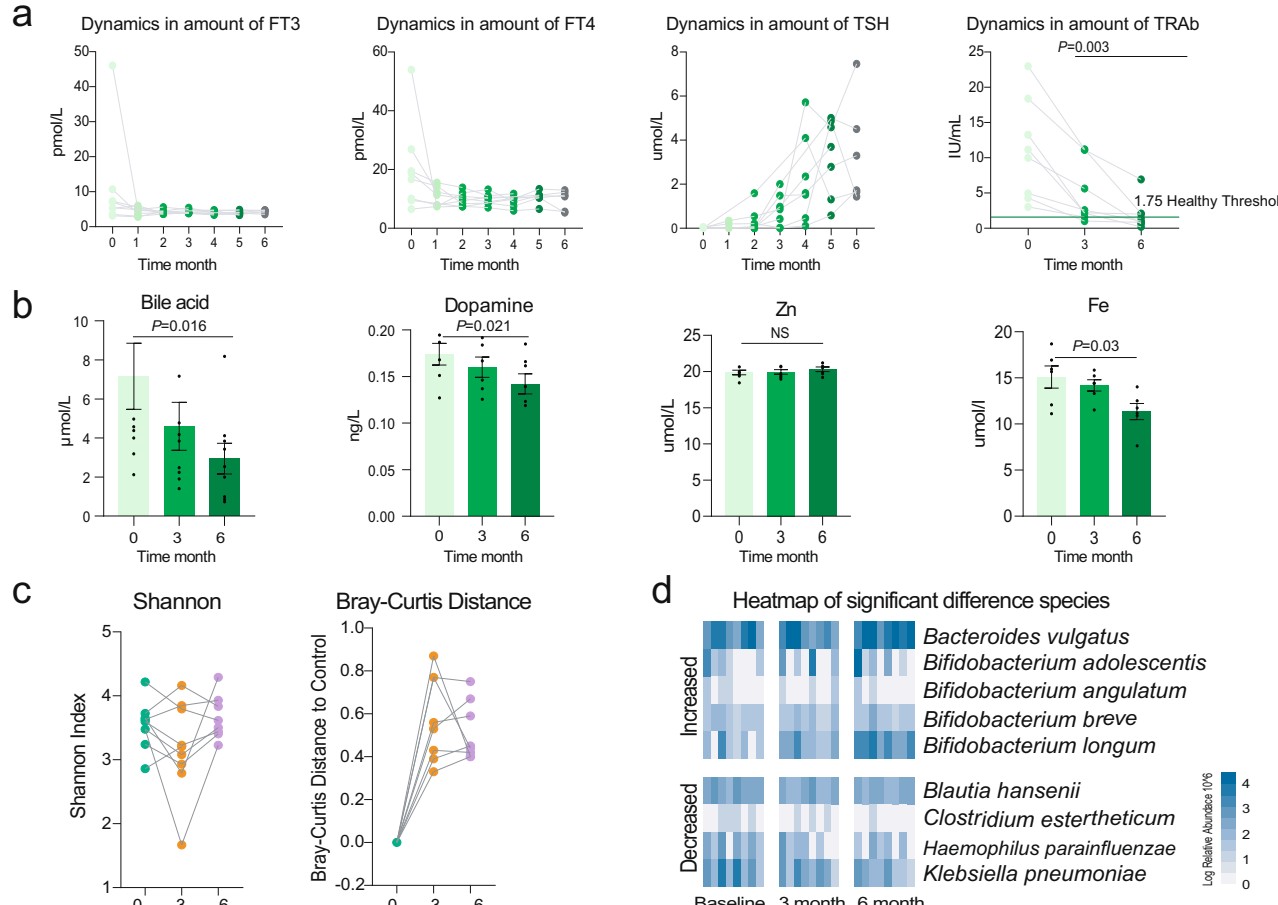

**Fig. 6 Probiotic Bifidobacterium longum adjuvant methimazole treatment improved the thyroid function and significantly reduced the recurrence rate of GD patients. a** Dynamic in the concentration of thyroid indexes including FT3, FT4, TSH, and TRAb during the whole experiment, these indexes were determined monthly. **b** Dynamic in the concentration of bloody bile acid, Dopamine, Zn, and Fe, these indexes were determined at baseline, month 3 and month 6. **c** The impacts of methimazole on intestinal microbial alpha diversity, the points in the same color represented the different subjects in the same time points. **d** The significantly different metagenomic species among different time points.

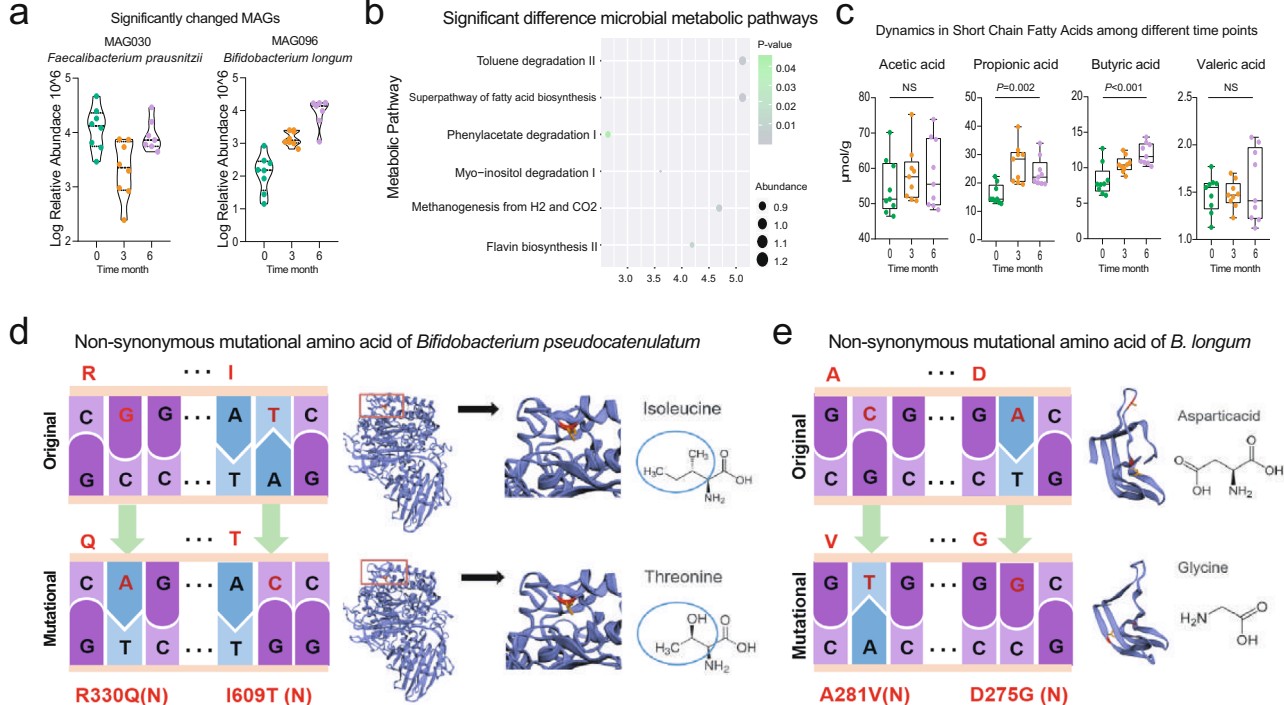

**Fig. 7 Probiotic Bifidobacterium longum adjuvant methimazole treatment improved intestinal microbial structure. a** MAGs of significant difference among different time points. **b** The intestinal microbial metabolic pathway that differed significantly between the samples at baseline and the samples with 6 months of probiotic and methimazole treatment. **c** Dynamics in Short-Chain Fatty Acids among different time points. **d**, **e** The evolutionary changes at the genetic level in intestinal microbial species that responded to probiotics intake. **d** Two SNPs (nonsynonymous mutations) were annotated at the *Bifidobacterium pseudocatenulatum* genome and the mutational gene was related to the function of Ig-like domain-containing protein. **e** The other two SNPs (nonsynonymous mutations) were annotated at the *Bifidobacterium longum* genome and the mutational gene was related to the function of LPXTG cell wall anchor domain-containing protein.

function and reduced the concentration of TRAb, which was an indicator of GD relapse rates.

## Discussion

Graves' disease (GD) is an autoimmune disorder that frequently results in hyperthyroidism and other symptoms, such as irritability, insomnia, diarrhoea, and weight loss[3]. In this study, we reported the curative effects of different treatment groups, including the methimazole (MI) treatment group, the MI + black bean treatment group (GB), and the MI + probiotic (*Bi®dobacterium longum*) treatment group, on thyroid function improvement in patients with GD, as well as the responses of host indigenous microbiota to different treatments at the ecological and evolutionary scales. Unsurprisingly, methimazole intake sharply decreased thyroid indexes, such as FT3, FT4, and TSH within just one month. However, we also observed a dramatic response of indigenous microbiota to methimazole intake, which implied that traditional GD therapy drugs significantly changed the patients' intestinal microbiome. Furthermore, these disturbances were reflected at the intestinal microbial taxonomic and metabolic pathway levels, even leading to microbial genomic mutations. Few studies have reported similar results at the intestinal microbiome scale[23], which also gives us more consideration in terms of traditional medicine therapy for patients with GD. The phenomenon was more similar to the widely used antibiotics that are a double-edged sword. Antibiotics are, indeed, not only longer considered only beneficial but also potentially harmful drugs, as their abuse appears to play a role in the pathogenesis of several disorders associated with the microbiota disorder[24]. Accordingly, the benefits brought by traditional medicine may be short-term gains but could not fundamentally

cure the disease[25]. Looking back at the present study, we also observed that the average amounts of thyrotropin receptor antibody (TRAb) at month 6 were still well above the healthy control level, which indicated high GD relapse rates in this treatment group (Fig. 2a). Overall, when the traditional GD medicine methimazole was applied for thyroid disease treatment, we still needed to rethink adjuvant therapy from the perspective of intestinal microecology.

To address the challenges mentioned above, we tried to supply the widely used microecological regulator probiotic with methimazole for GD treatment. Compared with the methimazole-only treatment group, we found that the SCFAs represented intestinal beneficial microbial metabolites that increased significantly in the probiotic group, while the concentrations of blood Fe, bile acid, and dopamine decreased sharply. Amazingly, we observed that TRAb in the probiotic group recovered to the healthy control level, which implied that probiotic adjuvant methimazole treatment improved thyroid function of patients with GD. Iron (Fe) and zinc (Zn) are minerals that are reported to support thyroid function. It has been shown that thyroid dysfunction is linked to abnormal levels of these minerals[26]. Studies have reported that mothers with goiters have lower iodine, Se, and Fe serum levels than healthy controls[27]. Bile acids are able to regulate energy metabolism through changes in TSH levels, and the total bile acid levels in the blood are decreased in patients with subclinical hypothyroidism[28]. Accordingly, the significantly changed concentrations of blood Fe and bile acid in the patients with GD of the probiotic group were closely related to thyroid function improvement. Furthermore, recent studies have highlighted the essential role of the gut microbiome in maintaining the immune system and human health[29]. GD-represented autoimmune thyroid disease is the most prevalent organ-specific autoimmune disease.

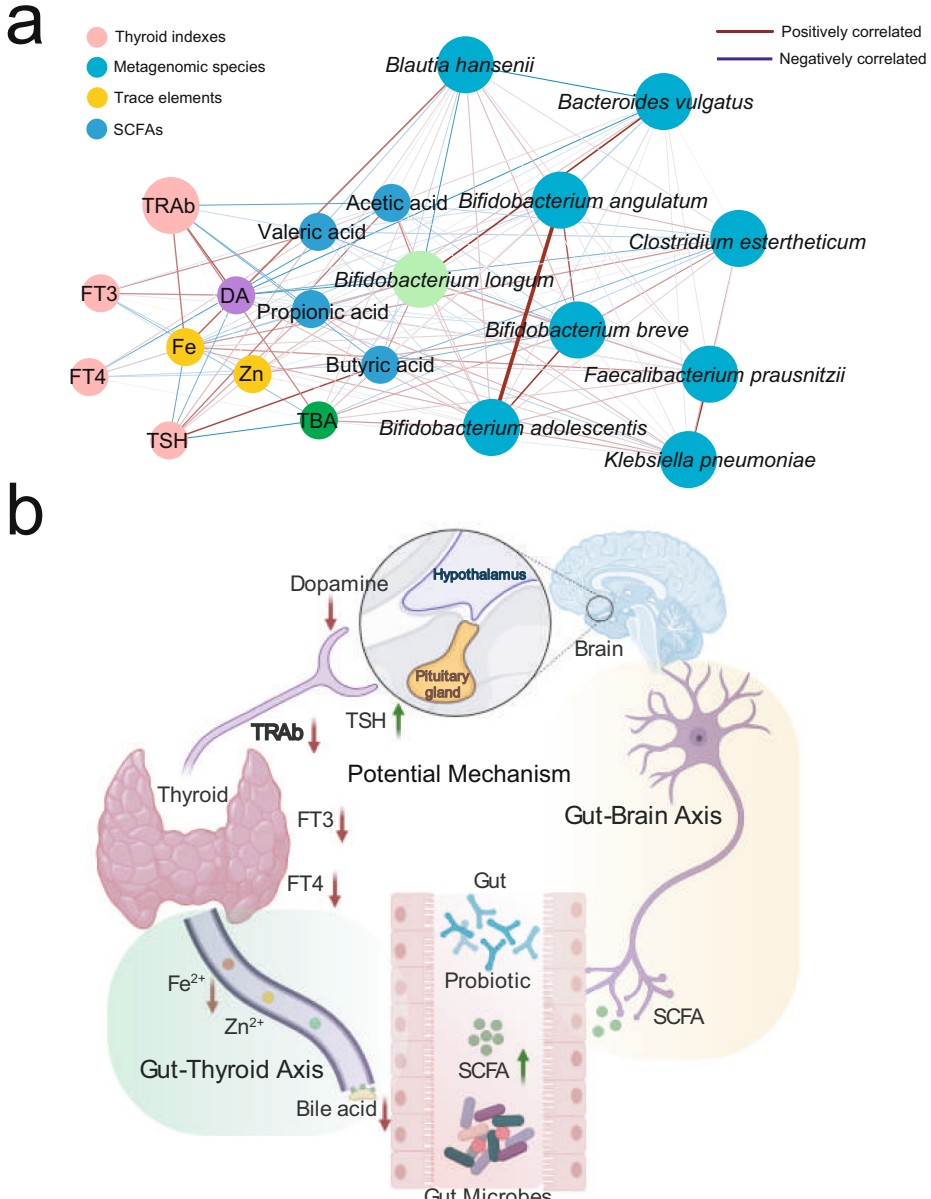

**Fig. 8 The potential mechanism underlying the interaction between probiotics and thyroid clinical indexes. a** The network including probiotic *Bifidobacterium longum*, probiotic closely related metagenomic species, SCFAs, dopamine, bile acid, bloody trace elements, and thyroid functional indexes based on the determined Spearman's rank correlation coefficients. The edge widths and colors (red: positive correlated and blue: negative correlated) are proportional to the correlation strength of co-occurrence probability. The node sizes are proportional to the mean abundance in the respective population. **b** The simplified diagrammatic drawing, including the gut-brain axis and gut-thyroid axis to visualize the potential effective mechanism.

Altered microbiota composition in the gut, as well as decreased microbial products, particularly short-chain fatty acids (SCFAs), promotes the development of autoimmune thyroid disease by several hypothesized mechanisms, including controlling the integrity of intercellular junctions and microbial transcriptomic, proteomic, and metabolic changes[19]. Here, the consumed probiotics regulated the intestinal microbiota and promoted the secretion of SCFAs. The beneficial microbial metabolite SCFAs influence host neurotransmitters, such as dopamine, in the brain and regulate the hypothalamus-pituitary axis (HPA)[30]. Dopamine inhibited the activity of the anterior pituitary gland, which may also regulate the thyroid indexes of GD patients[31]. Taken together, the potential effective mechanism could be attributed to two axes, the gut-brain

axis and the gut-thyroid axis, which finally improved the host's thyroid function and reduced the TRAb concentration, which is an indicator of the GD recurrence rate.

The limitation of the present study, including the following two points. On one hand, the sample size of each group was still small, many subjects at the baseline dropped due to the long experiment period and other various reasons. On the other hand, we did not track the patient's clinical and thyroid indexes after the 6 months experiment, which greatly limited the deduction of probiotic reducing the GD recurrence rate in our present research.

## Materials and methods

**Experimental design and subject recruitment**. In the present study, all GD patients were recruited from the Hainan Provincial People's Hospital, Haikou,

China. The subjects' basic information (gender, age, BMI, smoking, and alcoholism) and clinical indexes were recorded in Supplementary Data 1 and Supplementary Data 2. They were divided into three groups including the methimazole treatment group (GA, $n = 8$, 20 mg methimazole per day (Thyrozol tablets)), the methimazole + black-bean treatment group (GB, $n = 9$, 20 mg methimazole per day (Thyrozol tablets) and 100 g black bean per day), and the methimazole + probiotic (*Bifidobacterium longum*) treatment group (GC, $n = 9$, 20 mg methimazole per day (Thyrozol tablets) and $2 \times 10^7$ CFU per day), all subjects stayed on their treatment for 6 months (Fig. 1).

The study was reviewed and approved by the Ethics Committee of the Hainan Provincial People's Hospital (2018-109), and informed consent was obtained from all volunteers in written form before they were enrolled in the study. Sampling and all described subsequent steps were conducted in accordance with the approved guidelines. Blood samples were collected monthly, while the fecal samples were collected from each subject at baseline, 3 months and 6 months after treatment. For each Graves' disease patient, their fecal and blood samples were collected by a doctor during their clinical visit. After the weight of the fecal materials was determined, a sample protector (CW0592M, CWBIO, China) was added at a ratio of five-to-one to the sample to stabilize nucleotides. The samples were stored at −20 °C until further processing.

**Clinical indexes and short-chain fatty acids determination**. The thyroid indexes including free triiodothyronine (FT3), free thyroxine (FT4), thyroid-stimulating hormone (TSH), and thyroid-stimulating hormone receptor antibodies (TRAb), bloody trace elements, including Fe and Zn, dopamine, and bile acid were determined by using the enzyme-linked immunosorbent assay (ELISA) method. As mentioned before, the SCFAs in the gut, which included acetic acid, propionic acid, butyric acid, and valeric acid, were analyzed by the gas chromatography-mass spectrometry (GC-MS) described as previous studies[32].

**Fecal DNA extraction, shotgun metagenomic sequencing, and data quality control**. The QIAamp® DNA Stool Mini Kit (Qiagen, Hilden, Germany) was used for DNA extraction from the fecal samples. The quality of the extracted DNA was assessed by 0.8% agarose gel electrophoresis, and the OD 260/280 was measured by spectrophotometry. All of the DNA samples were subjected to shotgun metagenomic sequencing by using an Illumina HiSeq 2500 instrument in the Novogene Company (Beijing, China). Libraries were prepared with a fragment length of approximately 300 bp. Paired-end reads were generated using 100 bp in the forward and reverse directions. The quality of the reads was controlled by FastQC and was subsequently aligned to the human genome to remove the host DNA fragments.

Identification of microbial species, functional genes, and metabolic pathways The shotgun reads were assembled into contigs and scaffolds using MEGAHIT[33] with the default parameters. For metagenomic species annotation, the Bracken software was applied[34]. For metagenomic functional features and metabolic pathway annotation, HUMAnN2[35] was performed by using the UniRef90 database[35]. More information was listed in "code availability". Accordingly, we got the relative abundance of intestinal microbial taxonomic, gene families, and metabolic pathway profiles.

**Intestinal microbial antibiotic resistance genes and mobile elements profiles construction**. Intestinal microbial resistome and mobilome were characterized by mapping metagenomic reads to a comprehensive non-redundant database of more than 2700 mobile genetic elements and 3100 antibiotic resistance genes reported in the previous research[36] by Bowtie2 with options -D 20 -R 3 -N 1 -L 20 -i S,1,0.50. SAMtools software was used to filter and count reads and if both the reads mapped to the same gene the read was counted as one match and if the reads mapped to different genes, both were counted as hits to the respective gene.

**Construction of metagenome-assembled genomes (MAGs) and reconstruction of a phylogenomic tree of MAGs**. For metagenomic species analysis, MetaBAT[37] was performed to construct the MAGs by binning shotgun reads. After binning, the MAGs were assigned to a given reference genome if more than 80% of the subgene identified by Prodigal matched the same genome using BLASTn at a threshold of 95% identity over 90% of the gene length. Overall, the GTDB-Tk (V1.40) software was applied for MAGs taxonomic annotation and phylogenetical tree construction[38]. Specially, the parameter in the present study was used as follow: 1-For taxonomic assigned: gtdbtk classify_wf --genome_dir /uniq_mag/ --out_dir / mag_anotation --cpus 32 --extension fa –force; 2-For phylogenetic tree construction: time gtdbtk infer --msa_file /gtdbtk.bac120.user_msa.fasta --out_dir infer --cpus 8. The resulting phylogenetic tree was visualized using iTOL v4[39].

**Evolutionary analysis based on shotgun metagenomic data of gut microbiota**. We employed the MIDAS (Metagenomic Intra-Species Diversity Analysis System) to perform intestinal microbiota mutations annotation[40]. Briefly, a reference genome database including 34 species with the abundance more than 0.1% was constructed. Then the shotgun metagenomic sequencing reads were mapped to the database for intestinal species SNP calling. An SNP was confirmed only when: i. the mutation was detected compared with the original base of the microbial genome at

baseline; ii. the mutation was detectable in the following two time points; iii. the quantity score for each annotated SNP was more than 60. The SNPs profiles for these intestinal microbes were deposited in GitHub: https://github.com/zhjch321123/GD_Probiotic_treatment.

**Statistics statement**. All statistical analyses were performed using the R software (v3.5.1). PCoA analysis was performed in R using the "ade4" package (v1.7). CLR transformation was performed by the "zCompositions" package (v1.3.2). Heatmaps were constructed using the "pheatmap" package (v1.0.12). The differential abundances of various profiles were tested with the Wilcoxon rank-sum test and were considered significantly different at $p < 0.05$. For boxplot construction, the package "ggpubr" was used (v0.2.3). The edges of the network were calculated based on the determined Spearman's rank correlation coefficients and visualized in Cytoscape v3.7.1[41]. The potential mechanism figure was constructed by using an online tool named "BioRender".

**Reporting summary**. Further information on research design is available in the Nature Research Reporting Summary linked to this article.

## Data availability
The sequence data reported in this paper have been deposited in the NCBI database (resequencing and metagenomic sequencing data: PRJNA693409).

## Code availability
All the project analysis code had been deposited in GitHub: https://github.com/zhjch321123/GD_Probiotic_treatment.

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

## Acknowledgements

This research was supported by the Key Research and Development Project of Hainan Province (No. ZDYF2018111 and ZDYF2019150).

## Author contributions

The research topic was developed by J.Z., Y.P., and K.C. The experiment was performed by H.C., Q.O., and D.H. Data collection was performed by K.C. and C.C. Data analysis wee performed by D.H., S.J., and J.Z. The manuscript was written and revised by D.H., C.C., and J.Z. All authors read and approved the final manuscript for submission.

## Competing interests

The authors declare no competing interests.
