## [Peer Review File · Communications Biology]

Reviewers' comments:

Reviewer #1 (Remarks to the Author):

The authors report an interesting study on the impact of methimazole (MMI) alone or in combination with black bean or probiotic, on clinical outcomes in Graves' disease. This reviewer was unable to locate table 1, which details the patient characteristics of the 3 groups, in any of the online material.

Without this information it is not possible to evaluate the work.

Of note, the study comprises very small numbers of patients in each group (n= 8 or 9) and subjects have been followed for only 6 months following treatment which is too short considering that relapse can occur once MMI is withdrawn.

Reviewer #2 (Remarks to the Author):

Title: Probiotic *Bifidobacterium longum* supplied with methimazole reduced the relapse rates of Graves' disease through the gut-thyroid axis

Graves' disease is the most common systemic autoimmune disorder disease, current medication such as MMI treatment is strongly associated with a high risk of recurrence. The authors claimed the consumed probiotic regulated the intestinal microbiota and metabolites. These metabolites impacted neurotransmitter and blood trace elements through the gut-brain axis and gut-thyroid axis, which finally improved the host's thyroid function and reduced the relapse rates of GD. As such, the article is original and timely, taking into account that we need a better understanding of what is happening when a GD patient takes probiotics. However, the manuscript needs some minor revision and clarification.

1. Please add the sample size in the abstract section.
2. Page 3, Line 64-70, divided the long sentence into 2-3 short sentences for clarification.
3. Line 102, "Then, a sharp decline was observed in the microbial Shannon index", the statistical difference should be marked in Figure 2C.
4. Line 110, the *Lactobacillus salivarius* was renamed as *Ligilactobacillus salivarius*
5. Line 132, the methods for SNPs identification should be described in details.
6. Why do the authors choose Black Bean for GD treatment?
7. The software version should be clarified in the "Statistics statement" section.
8. The methods for MAG construction should be described in details.

Reviewer #3 (Remarks to the Author):

The submitted manuscript describes effects of methimazole alone and in combination with either black bean or probiotic *Bifidobacterium longum* on microbiota diversity, metabolites and thyroid hormone levels. The data suggest that administration of probiotics can decrease TRAb levels and increase TSH levels. The addition of probiotics may be an interesting option to increase the efficacy of methimazole and prevent relapse. The manuscript in general is clearly written but information on patients has to be added and several questions answered.

Comments

- The description of the patient population lacks information, such as duration (pretreatment?) or newly diagnosed Graves' disease, serum parameters, particularly for indication of liver damage (e.g. ALT, AP, bilirubin) at later time points.
- Further, indication of dose and of product names of the medication has to be added.
- What is the motivation to use black bean as control?
- Relapse was associated to T3/FT4 ratios, are these data available (only FT3 is shown) for the patients?
- Methimazole is poorly metabolized in the human body and undergoes enterohepatic circulation.

Do the authors suspected contribution of different extent of metabolisation by the bacteria?

-The influence on actual relapse is not known since the study was finished at 6 month. Are there any follow up data available?

-The limitation of low sample number and also the possibility that this is only transient should be mentioned.

-The authors speculate on effects on the gut-brain axis. Could they elaborate a bit on that because dopamine cannot cross the blood-brain barrier? Was there a link between blood dopamine and TSH levels in individual patients? The link of TSH levels to dopamine (the hypothesis if the gut-brain axis) is not convincing as TSH levels in the bean and control groups in general were higher than in the probiota group. For a better comparison, it would have been better to use the same scale as for most of the other diagrams.

Comments for Manuscript number: COMMSBIO-21-0922

Title: " Probiotic *Bifidobacterium longum* supplied with methimazole reduced the relapse rates of Graves' disease through the gut-thyroid axis "

Reviewers' comments:

Reviewer #1 (Remarks to the Author):

The authors report an interesting study on the impact of methimazole (MMI) alone or in combination with black bean or probiotic, on clinical outcomes in Graves' disease.

This reviewer was unable to locate table 1, which details the patient characteristics of the 3 groups, in any of the online material.

Without this information it is not possible to evaluate the work.

Response: We apologized for the missing supplemental tables including the subjects' information and the clinical indexes. Accordingly, we have reformatted the Table S1 and Table S2 in the "Office Word" documents.

Of note, the study comprises very small numbers of patients in each group (n= 8 or 9) and subjects have been followed for only 6 months following treatment which is too short considering that relapse can occur once MMI is withdrawn.

Response: Thank you for your concern. Yes, we have to admit the sample size of each group was still small. Actually, there were more than 20 subjects in each group at the baseline. However, during the 6 months experiment, many subjects dropped due to various reasons. We sincerely wish the reviewer could understand the difficulties in subjects recruiting and successive sample collection in our present study.

Meanwhile, we deeply agree with your concern in the overstatement of the GD relapse rate. Since we do not have any follow up data after 6 months, we can hardly speculate the probiotic consumption reduced the relapse rate of GD patients. To avoid the overstatement, we decide to tune down our claims in the revised manuscript. So, we have deleted the statement of "reduced the GD recurrence rate", just reported the probiotic intake improved the patients' thyroid function. At last, we added some sentences to point out the limitation of the present study at the end of discussion section as follow, which should be addressed in our future research.

"The limitation of the present study including the following two points. On one hand, the sample size of each group was still small, many subjects at the baseline dropped due to the long experiment period and other various reasons. On the other hand, we did not track the patient's clinical and thyroid indexes after the 6 months experiment, which greatly limited the deduction of probiotic reducing the GD recurrence rate in our present research. "

Reviewer #2 (Remarks to the Author):

Title: Probiotic *Bifidobacterium longum* supplied with methimazole reduced the relapse rates of Graves' disease through the gut-thyroid axis

Graves' disease is the most common systemic autoimmune disorder disease, current medication such as MMI treatment is strongly associated with a high risk of recurrence. The authors claimed the consumed probiotic regulated the intestinal microbiota and metabolites. These metabolites impacted neurotransmitter and blood trace elements through the gut-brain axis and gut-thyroid axis, which finally improved the host's thyroid function and reduced the relapse rates of GD. As

such, the article is original and timely, taking into account that we need a better understanding of what is happening when a GD patient takes probiotics. However, the manuscript needs some minor revision and clarification.

Response: We appreciate the reviewer's insightful comments which allowed us to improve the manuscript. Please find our point-to-point responses below.

1. Please added the sample size in the abstract section.

Response: Thanks, we have added the sample size in the abstract section.

2. Page 3, Line 64-70, divided the long sentence in to 2-3 short sentences for clarification.

Response: Thank you for raising this point. We have rewritten the long sentence as follow: "Probiotics are live microorganisms that can potentiate health benefits on hosts when administered in adequate amounts. In general, the probiotics contained *Bifidobacterium* spp. and *Lactobacillus* spp., which are well known for their role in regulating and rebuilding the host's gut microbiome. Recent studies reported the probiotics had been used to prevent and treat a variety of metabolic diseases, including type 2 diabetes, polycystic ovary syndrome and hyperglycaemia."

3. Line 102, "Then, a sharp decline was observed in the microbial Shannon index", the statistical difference should be marked in Figure 2C.

Response: Thanks for your concern, we have modified this point per your suggestion.

4. Line 110, the *Lactobacillus salivarius* was renamed as *Ligilactobacillus salivarius*

Response: Thanks for your concern, we have modified this point per your suggestion.

5. Line 132, the methods for SNPs identification should be described in details.

Response: Thank you for your comment, we have described the method for SNP identification in more detail in line xx to xx as follow:

"We aligned the metagenomic data against the reference genomes of species with relative abundances higher than 0.5% in the subjects of group A at different time points and reconstructed a profile of SNPs (a SNP was identified and confirmed only when the following occurred: *i.* the mutation was detected compared with the original base of the microbial genome at baseline; *ii.* the mutation was detectable in the following two time points; *iii.* the quantity score for each annotated SNP was more than 60)."

6. Why the authors choose Black Bean for GD treatment?

Response: Thanks for your concern. Graves' disease is a consumptive disease, so the patient should be arranged a nutrient-rich, easily digestible diet, supplemented with enough calories, nutrients and vitamins to correct the consumption caused by the disease. In traditional Chinese medicine, black beans can be used as a dietary prescription for hyperthyroidism alone, which has a good effect on improving the symptoms of hyperthyroidism patients, such as post-illness weakness and excessive sweating [1,2]. At the same time, black beans contain rich protein, fat, carbohydrates, vitamins and a variety of minerals. Black bean has high energy and is easy to digest, which is of great significance to meet the metabolic consumption needs of patients with Graves' disease.

Meanwhile, the statement above had been added into the Introduction section of the revised manuscript.

[1] Dianyun Li, Jian Li. *Selection of pharmacological diet prescription for hyperthyroidism. Oriental Medicinal Food.* 2007, (11).

[2] Junli Lin. *Medicinal diet therapy for hyperthyroidism. Chinese Folk Therapy.* 2004, (12).

7. The software version should be clarified in the “Statistics statement” section.

Response: Thanks, we have added the software version in the “Statistics statement” section per your comment.

8. The methods for MAG construction should be described in details.

Response: Thanks for your concern, we have described the method for MAG assembling in more detail in line xx to xx as follow:

“For metagenomic species analysis, MetaBAT was performed to construct the MAGs by binning shotgun reads. After binning, the MAGs were assigned to a given reference genome if more than 80% of the sub-gene identified by Prodigal matched the same genome using BLASTn at a threshold of 95% identity over 90% of the gene length. Overall, the GTDB-Tk (V1.40) software was applied for MAGs taxonomic annotation and phylogenetical tree construction. Specially, the parameter in the present study was used as follow: 1-For taxonomic assigned: `gtdbtk classify_wf --genome_dir /uniq_mag/ --out_dir / mag_annotatation --cpus 32 --extension fa --force`; 2-For phylogenetic tree construction: `time gtdbtk infer --msa_file /gtdbtk.bac120.user_msa.fasta --out_dir infer --cpus 8`. The resulting phylogenetic tree was visualized using iTOL v4.”

Reviewer #3 (Remarks to the Author):

The submitted manuscript describes effects of methimazole alone and in combination with either black bean or probiotic *Bifidobacterium longum* on microbiota diversity, metabolites and thyroid hormone levels. The data suggest that administration of probiotics can decrease TRAb levels and increase TSH levels. The addition of probiotics may be an interesting option to increase the efficacy of methimazole and prevent relapse. The manuscript in general is clearly written but information on patients has to be added and several questions answered.

Response: We appreciate the reviewer’s insightful comments which allowed us to improve the manuscript. Please find our point-to-point responses below.

Comments

-The description of the patient population lacks information, such as duration (pretreatment?) or newly diagnosed Graves’ disease, serum parameters, particularly for indication of liver damage (e.g. ALT, AP, bilirubin) at later time points.

Response: We apologized for the missing supplemental tables including the subjects’ information and the clinical indexes. Accordingly, we have reformatted the Table S1 and Table S2 in the “Office Word” documents.

-Further, indication of dose and of product names of the medication has to be added.

Response: Thank you for your concern, these information had been added in the revised manuscript as follow: “They were divided into 3 groups including the methimazole treatment group (GA, $n=8$, 20 mg methimazole per day (Thyrozol tablets)), the methimazole+black-bean treatment group (GB, $n=9$, 20 mg methimazole per day (Thyrozol tablets) and 100 g black-bean per day), and the methimazole+probiotic (*Bifidobacterium longum*) treatment group (GC, $n=9$, 20 mg methimazole per day (Thyrozol tablets) and 2×10^7 CFU per day), all subjects stayed on their treatment for 6 months.”

-What is the motivation to use black bean as control?

Response: Thanks for your query. Graves' disease is a consumptive disease, so the patient should be arranged a nutrient-rich, easily digestible diet, supplemented with enough calories, nutrients and vitamins to correct the consumption caused by the disease. In traditional Chinese medicine, black beans can be used as a dietary prescription for hyperthyroidism alone, which has a good effect on improving the symptoms of hyperthyroidism patients, such as post-illness weakness and excessive sweating [1,2]. At the same time, black beans contain rich protein, fat, carbohydrates, vitamins and a variety of minerals. Black bean has high energy and is easy to digest, which is of great significance to meet the metabolic consumption needs of patients with Graves' disease. Meanwhile, the statement above had been added into the Introduction section of the revised manuscript.

[1] Dianyun Li, Jian Li. *Selection of pharmacological diet prescription for hyperthyroidism. Oriental Medicinal Food. 2007, (11).*

[2] Junli Lin. *Medicinal diet therapy for hyperthyroidism. Chinese Folk Therapy. 2004, (12).*

-Relapse was associated to T3/FT4 ratios, are these data available (only FT3 is shown) for the patients?

Response: Thank you for your query, we did not determine the T3 indexes of GD patients, but exhibited the FT4 data in supplementary table 2. In clinical, the thyrotropin receptor antibody (TRAb) was also an indicator of the GD recurrence rate.

-Methimazole is poorly metabolized in the human body and undergoes enterohepatic circulation. Do the authors suspected contribution of different extent of metabolisation by the bacteria?

Response: We appreciate your very insightful concern. Drug metabolism by gut bacteria is the frontiers in gut microbiome research. The gut microbiota is implicated in the metabolism of many medical drugs, with consequences for interpersonal variation in drug efficacy and toxicity. However, quantifying microbial contributions to drug metabolism in vivo is challenging, particularly in cases where host and microbiome perform the same metabolic transformation [3]. Although, Goodman did not answer the question which bacteria can metabolize Methimazole through comprehensive drug metabolism experiments [4]. They discovered that, for two thirds (176/271) of the assayed drugs, the level of the drug after incubation was significantly reduced by at least one bacterial strain, and that each strain metabolizes 11–95 drugs. Donia also identified 57 drugs as microbiome-derived metabolism (MDM)-Screen by testing a diverse library of 575 orally administered drugs [5]. Unfortunately, the evidence of gut microbes metabolizing Methimazole is limited. We hold the view that individual differences in the gut microbiome may reflect

differences in the ability to metabolize Methimazole. Also, it may be difficult to answer the scientific question of the contribution of gut microbes to drug metabolism in the absence of results based on isolation and culture, and germ-free mice, which is still a gap should be further explored.

[3] Michael Zimmermann, Maria Zimmermann-Kogadeeva, Rebekka Wegmann, Andrew L. Goodman. *Separating host and microbiome contributions to drug pharmacokinetics and toxicity*[J]. *Science*,2019,363(6427).

[4] Michael Zimmermann, Maria Zimmermann-Kogadeeva, Rebekka Wegmann, Andrew L. Goodman. *Mapping human microbiome drug metabolism by gut bacteria and their genes*[J]. *Nature: International weekly journal of science*,2019,570(7762).

[5] Bahar Javdan, Jaime G. Lopez, Pranatchareeya Chankhamjon, Ying-Chiang J. Lee, Raphaella Hull, Qihao Wu, Xiaojuan Wang, Seema Chatterjee, Mohamed S. Donia. *Personalized Mapping of Drug Metabolism by the Human Gut Microbiome*[J]. *Cell*,2020,181(7).

-The influence on actual relapse is not known since the study was finished at 6 month. Are there any follow up data available?

Response: Thank you for your concern, we deeply agree with your comment. Since we do not have any follow up data after 6 months, we can hardly speculate the probiotic consumption reduced the relapse rate of GD patients. To avoid the overstatement, we decide to tuned down our claims in the revised manuscript. So, we have deleted the statement of “reduced the GD recurrence rate”, just reported the probiotic intake improved the patients’ thyroid function.

-The limitation of low sample number and also the possibility that this is only transient should be mentioned.

Response: Thank you for your insightful comment. Yes, we have to admit the sample size of each group was still small. Actually, there were more than 20 subjects in each group at the baseline. However, during the 6 months experiment, many subjects dropped due to various reasons. We sincerely wish the reviewer could understand the difficulties in subjects recruiting and successive sample collection in our present study.

Meanwhile, we added some sentences to point out the limitation of the present study at the end of discussion section as follow, which should be addressed in our future research.

“The limitation of the present study including the following two points. On one hand, the sample size of each group was still small, many subjects at the baseline dropped due to the long experiment period and other various reasons. On the other hand, we did not track the patient’s clinical and thyroid indexes after the 6 months experiment, which greatly limited the deduction of probiotic reducing the GD recurrence rate in our present research. “

-The authors speculate on effects on the gut-brain axis. Could they elaborate a bit on that because dopamine cannot cross the blood-brain barrier? Was there a link between blood dopamine and TSH levels in individual patients? The link of TSH levels to dopamine (the hypothesis if the gut-brain axis) is not convincing as TSH levels in the bean and control groups in general were higher than in the probiotic group. For a better comparison, it would have been better to use the same scale as for most of the other diagrams.

Response: We appreciate your insightful comment and acute insight! After discussing with my

team, we also found the negative correlation between the dopamine and TSH levels was overstated. As your comment, the TSH level in the bean and control groups in general were higher than in the probiotic group, but we did not observe any significant difference in the dopamine level in the bean and control groups. Accordingly, we have deleted the statement related to the link of TSH levels to dopamine in the revised manuscript.

Even so, we could not ignore the potential role of gut-brain axis because of the dynamic in SCFAs level in the probiotic group. It was convincing that the consumed probiotics regulated the intestinal microbiota and promoted the secretion of SCFAs. The beneficial microbial metabolite SCFAs influence host neurotransmitters, such as dopamine, in the brain and regulate the hypothalamus-pituitary axis (HPA) [6]. Dopamine inhibited the activity of the anterior pituitary gland, which may also regulate the thyroid indexes of GD patients. [7] So, we would like to retain the gut-brain axis in the potential mechanism figure.

[6] Farzi, A., Frohlich, E. E. & Holzer, P. *Gut Microbiota and the Neuroendocrine System. Neurotherapeutics* 15, 5-22, doi:10.1007/s13311-017-0600-5 (2018).

[7] Neuman, H., Debelius, J. W., Knight, R. & Koren, O. *Microbial endocrinology: the interplay between the microbiota and the endocrine system. FEMS Microbiol Rev* 39, 509-521, doi:10.1093/femsre/fuu010 (2015).

REVIEWERS' COMMENTS:

Reviewer #2 (Remarks to the Author):

All my concerns have been addressed accordingly.

Reviewer #3 (Remarks to the Author):

The authors addresssed my comments and I understand that specific parameters (e.g. group sizes) cannot be improved.